# An Improvement in Diagnostic Blood Culture Conditions Allows for the Rapid Detection and Isolation of the Slow Growing Pathogen *Yersinia pestis*

**DOI:** 10.3390/pathogens11020255

**Published:** 2022-02-16

**Authors:** Efi Makdasi, Yafit Atiya-Nasagi, David Gur, Ayelet Zauberman, Ofir Schuster, Itai Glinert, Shlomo Shmaya, Elad Milrot, Haim Levy, Shay Weiss, Theodor Chitlaru, Emanuelle Mamroud, Orly Laskar

**Affiliations:** Israel Institute for Biological Research, Ness-Ziona 74100, Israel; yafita@iibr.gov.il (Y.A.-N.); gurd@iibr.gov.il (D.G.); ayeletz@iibr.gov.il (A.Z.); ofirsc@iibr.gov.il (O.S.); itaig@iibr.gov.il (I.G.); shlomosh@iibr.gov.il (S.S.); eladm@iibr.gov.il (E.M.); haiml@iibr.gov.il (H.L.); shayw@iibr.gov.il (S.W.); theodorc@iibr.gov.il (T.C.); emmym@iibr.gov.il (E.M.)

**Keywords:** *Y. pestis*, plague, blood culture, F1 and V antigens, bacterial diagnostics

## Abstract

Plague, caused by the human pathogen *Yersinia pestis*, is a severe and rapidly progressing lethal disease that has caused millions of deaths globally throughout human history and still presents a significant public health concern, mainly in developing countries. Owing to the possibility of its malicious use as a bio-threat agent, *Y. pestis* is classified as a tier-1 select agent. The prompt administration of an effective antimicrobial therapy, essential for a favorable patient prognosis, requires early pathogen detection, identification and isolation. Although the disease rapidly progresses and the pathogen replicates at high rates within the host, *Y. pestis* exhibits a slow growth in vitro under routinely employed clinical culturing conditions, complicating the diagnosis and isolation. In the current study, the in vitro bacterial growth in blood cultures was accelerated by the addition of nutritional supplements. We report the ability of calcium (Ca^+2^)- and iron (Fe^+2^)-enriched aerobic blood culture media to expedite the growth of various virulent *Y. pestis* strains. Using a supplemented blood culture, a shortening of the doubling time from ~110 min to ~45 min could be achieved, resulting in increase of 5 order of magnitude in the bacterial loads within 24 h of incubation, consequently allowing the rapid detection and isolation of the slow growing *Y. pestis* bacteria. In addition, the aerobic and anaerobic blood culture bottles used in clinical set-up were compared for a *Y. pestis* culture in the presence of Ca^+2^ and Fe^+2^. The comparison established the superiority of the supplemented aerobic cultures for an early detection and achieved a significant increase in the yields of the pathogen. In line with the accelerated bacterial growth rates, the specific diagnostic markers F1 and LcrV (V) antigens could be directly detected significantly earlier. Downstream identification employing MALDI-TOF and immunofluorescence assays were performed directly from the inoculated supplemented blood culture, resulting in an increased sensitivity and without any detectable compromise of the accuracy of the antibiotic susceptibility testing (E-test), critical for subsequent successful therapeutic interventions.

## 1. Introduction

Plague, one of the oldest recorded human pestilences, is a severe and rapidly progressing lethal infectious disease that has caused millions of deaths throughout human history. The disease is still present in different parts of the world with the last 20 years witnessing a global rise in plague incidence. Recent outbreaks have been reported in Uganda [1], China [2], The Democratic Republic of Congo [3] and Madagascar [4].

*Y. pestis*, the causative agent of plague, is a Gram-negative bacterium, circulating among rodents and most commonly spread to humans via fleas. A delayed treatment results in high mortality rates [5]. Hence, the prompt identification of the pathogen is crucial for a proper and timely treatment.

According to the World Health Organization (WHO) indications, plague is diagnosed by the positive isolation and identification of *Y. pestis* in a culture or in retrospect by a seroconversion specific for the capsular F1 antigen.

Although the clinical course of the infection is characterized by a fast progression in vivo, all known *Y. pestis* strains exhibit slow a growth in vitro. A clinical diagnosis may require 24–72 h incubation to enable the colony isolation followed by an additional 24–48 h required for an accurate antibiotic susceptibility determination. The early administration of an effective antimicrobial therapy is essential for favorable patient outcomes; thus, the early detection and isolation of the pathogen is instrumental for an effective treatment. As bacteremia is common in bubonic, pneumonic and septicemic plague, clinical blood samples are relevant specimens for the isolation and diagnosis of *Y. pestis*. Specifically, blood cultures are considered to be the gold standard of sepsis diagnostics.

In clinical set-ups, peripheral blood sampling combined with automated alert culture systems are routinely implemented owing to their simplicity and high sensitivity. Yet, in more than 50% of patients, the bacteremia is less than 1 colony-forming unit (CFU)/mL [6]; therefore, a significant volume of blood sampling is necessary for detection (usually 10 mL per bottle in 2 to 3 sets of aerobic and anaerobic blood cultures). Taken together, the slow growth rate of *Y. pestis* and a low initial CFU count per sample potentially result in prolonged incubation times prior to a positive identification.

The importance of an early diagnosis has incented the development of culturing methods that exhibit an improved efficiency and selectivity for colony isolation [7,8,9] as well as rapid and sensitive approaches for the detection of *Y. pestis* directly from clinical samples. These are mainly based on immune-labeling [10,11], reporter phage [12,13,14] and genetic assays [15]. Usually, the selective media act by imposing a limitation of specific nutrients and/or the inclusion of selective antimicrobials, thus promoting the preferential growth of a particular targeted microorganism. Although effective in discerning amongst microorganisms, the restrictions imposed by the selective media tend to retard growth, often resulting in a delayed colony formation [7].

It is well-established that metal ions are essential elements in bacterial metabolism and growth. Specifically, iron (Fe^+2^) ions serve as a metabolic cofactor necessary for crucial processes including respiration, oxidative stress resistance and virulence factor production [16]. Generally, the level of iron required for an optimal bacterial growth is ~10^−6^ M; however, the level of free iron in mammalian tissues is typically ~10^−18^ M [17]. During an infection, the host immune response includes a further limitation of iron availability from the invading pathogens by a process termed nutritional immunity [18,19]. Enriching the iron availability in a culture by an active administration or by using hemolyzed human erythrocytes was reported to increase *Y. pestis* virulence [20,21,22]. Another nutrient demonstrated to be crucial for *Y. pestis* growth at 37 °C is calcium (Ca^+2^). Under low calcium conditions, the bacteria undergo a specific metabolic arrest termed a low calcium response (LCR), which is characterized by restricting the vegetative growth and accompanied by the expression of the Type III secretion system and the V antigen [23,24]. Adding blood-equivalent levels of Ca^+2^ (2.5 mM) results in a reversion of the response [25,26,27]. In spite of the effect of Fe^+2^ and Ca^+2^ ions on bacterial growth, their addition for expediting the culture time for diagnostic purposes was not reported.

In this study, we evaluated the bacterial growth rates of virulent *Y. pestis* strains in Fe^+2^- and Ca^+2^-enriched blood cultures as a means for improving and accelerating bacterial growth. The shortening of the culturing time facilitated a rapid pathogen detection and isolation, positively impacting on the clinical diagnosis and enabling the prompt onset of an efficient therapy.

## 2. Results and Discussion

### 2.1. Accelerated Growth Rates of Y. pestis in Iron- and Calcium-Supplemented Blood Cultures

In order to evaluate the bacterial growth rates of virulent *Y. pestis* strains in Fe^+2^- and Ca^+2^-enriched blood cultures, the Kimberley53 (Kim53, biovar Orientalis, Table 1) strain was spiked into BACTEC Plus Aerobic/F culture vials containing 10 mL of naïve fresh human blood. Initially, we assessed the impact of adding Fe^+2^ (100 µM) and Ca^+2^ (2.5 mM) separately or in a combination compared with a non-supplemented blood culture. The vials were incubated at 150 rpm at 37 °C for 24 h and the bacterial growth was determined during and following the incubation by colony-forming unit (CFU) counts. As shown in Figure 1, the blood cultures supplemented with Fe^+2^ and Ca^+2^ separately allowed increased *Y. pestis* growth rates compared with the non-supplemented media. Twenty-four hours after the initiation of the culture, the improved growth rate resulted in an increase of one order of magnitude in the bacterial counts. In addition, combining both supplements resulted in increase of 2order of magnitude in the bacterial counts following a 24 h incubation, suggesting an additive effect of the supplements. The bacterial doubling time showed a significant reduction from ~110 min in the non-supplemented blood culture to ~45 min in the combined Fe^+2^- and Ca^+2^-supplemented media. These results strengthened the conclusion that the co-administration of Ca^+2^ and Fe^+2^ to the blood culture vials was highly beneficial, potentially shortening the detection times for *Y. pestis* in a clinical setting.

The effect of Fe^+2^ and Ca^+2^ on the bacterial growth was further demonstrated in three additional *Y. pestis* strains, Alexander, IV 75 195 (biovar Orientalis, Table 1) and PKH-10 (biovar Medievalis, Table 1). As depicted in Figure 2, a significant increase in the bacterial counts was achieved in the supplemented samples (an increase of up to five orders of magnitude for all strains after an incubation of 24 h). It is important to note that in all tested strains, the bacterial counts in the supplemented media after 12 h growth were equivalent to or higher than those with a non-supplemented growth 24 h after inoculation. This finding, along with the shortening of the doubling time (Figure 1), could potentially shorten the duration of the positive growth signal in clinical samples, accelerating the identification of pathogens by downstream processing methods such as MALDI-TOF and immunoassays as well as isolation and culturing for phenotype characterization purposes such as antibiotic sensitivity tests.

Available Ca^+2^ and Fe^+2^ in blood cultures mainly originate from the components added by the manufacturer and the peripheral blood sample itself used for the inoculation of the vial in the course of clinical sampling. Although the effective concentration of Ca^+2^ (2.5 mM) required for an optimal *Y. pestis* growth was previously accurately determined [25,26,27], information regarding the Fe^+2^ levels enabling *Y. pestis* growth in blood cultures is limited. The low stability of iron together with concerns pertaining to the potential toxicity of high iron levels motivated the determination of the optimal concentration of iron in blood cultures for *Y. pestis* growth. Fe^+2^ (10–1000 µM) was administered together with a constant concentration of Ca^+2^ (2.5 mM) to aerobic blood culture vials and inoculated with either virulent *Y. pestis* Kim53 or PKH-10. The vials were incubated at 150 rpm at 37 °C for 24 h and the bacterial growth was determined during and following incubation by CFU counts. As shown in Figure 3, the growth rates of both strains were affected by the supplemented media, achieving an increase of three and four orders of magnitude in the Kim53 and PKH-10 counts, respectively. No significant differences in the bacterial counts were observed across the range of increased Fe^+2^ concentrations (10–1000 µM), indicating that 10 µM was a sufficient concentration for bacterial growth. The calculations of the bacterial doubling times under the supplemented conditions established a doubling time of ~45 min for both virulent strains compared with ~110 min (Kim53) and ~120 min (PKH-10) obtained in the non-supplemented media, substantiating the initial results. In spite of the observation that 10 µM Fe^+2^ was sufficient, considering the possible low stability on one hand and the potential toxicity of Fe^+2^ at high concentrations on the other hand, an Fe^+2^ concentration of 100 µM was employed in all subsequent experiments.

As diagnostic blood cultures are often carried out by the inoculation of BACTEC Standard Aerobic/F culture vials (already containing hemin as a component added by the manufacturer; see Appendix A) we determined whether the increase in the growth rate was manifested in these cultures as observed in the supplemented BACTEC Plus Aerobic/F cultures. To address this issue, Kim53 and PKH-10 strains were spiked into supplemented and non-supplemented BACTEC Standard Aerobic vials containing 10 mL of fresh blood and the bacterial growth was determined after 24 h as previously described. BACTEC Plus Aerobic/F culture vials were used as the positive control. As shown in Figure 4, a significant increase in the bacterial counts was observed in the supplemented BACTEC Standard Aerobic culture vials, similar to the bacterial growth quantified in the BACTEC Plus Aerobic/F vials. Moreover, no differences in the bacterial growth were observed in the non-supplemented vials irrespective of the components added by the manufacturer. Overall, these results stressed that the beneficial effect of supplemented Fe^+2^ and Ca^+2^ for increasing *Y. pestis* growth rates under aerobic conditions was manifested regardless of the BACTEC vial type used.

### 2.2. Y. pestis Growth in Aerobic and Anaerobic Blood Culture Vials

The current recommendations for blood culture sampling for adult patients consist of collecting at least two pairs of vials with each set requiring 20 mL of blood from two distinct sampling sites distributed equally between an aerobic and anaerobic vial [28,29]. Lately, the relevance of this routine has been challenged, mostly due to the fact that the yield of anaerobic bacteria is frequently limited in these vials and is not restricted to the isolation of strictly anaerobic microorganisms [30,31]. As the incidence of anaerobic bacteremia is low, anaerobic blood cultures tend to be selectively used in patients at risk of anaerobic infections.

In spite of the preferential clinical use of aerobic conditions, anaerobic blood culture protocols—which allow the growth of facultative organisms—may exhibit a superior sensitivity. As *Y. pestis* is classified as an anaerobic facultative bacterium, its growth in anaerobic vs. aerobic blood cultures was evaluated. Kim53 and PKH-10 bacteria were spiked into four different non-supplemented BACTEC blood cultures of two aerobic and two anaerobic vials. The vials were incubated at 150 rpm at 37 °C for 24 h and the bacterial growth was determined by the CFU counts in the course as well as following the incubation. The growth of the virulent Kim53 and PKH-10 strains was accelerated in the anaerobic vials. This was observed for the two strains both in standard and lytic anaerobic vials (Figure 5A,B). These results were in line with those reported for the attenuated *Y. pestis* EV76 using anaerobic bottles [32]. Of note, the current results strongly supported previous reports that suggested the utility of anaerobic culture conditions in the diagnosis of other blood stream infections (BSI) caused by facultative anaerobes, obligate anaerobic pathogens and even obligate aerobic pathogens [33,34,35]. Although this later observation may stem from the inadvertent introduction of air into the vials and/or to the rich composition of the medium [36], it may be concluded that the implementation of anaerobic growth conditions in the clinical diagnostic practice in general and specifically for *Y. pestis* is highly beneficial.

To evaluate whether the accelerated growth in the anaerobic vials was due to the anaerobic conditions or to the different composition of the media (see Appendix A), experiments on Kim53 bacteria-spiked standard and lytic anaerobic cultures were performed in the presence of exogenously added air, replacing the original N_2_/CO_2_ atmosphere of the cultures. A significant decrease in the bacterial counts was observed in these modified standard anaerobic vials compared with the non-altered vials (Figure 5C). Interestingly, the bacterial counts in the modified anaerobic culture were comparable with those observed in the Aerobic Plus blood cultures, indicating the importance of anaerobic conditions in accelerating the *Y. pestis* growth rates. In contrast, no effect from the addition of air into the lytic anaerobic cultures was observed, suggesting that the composition was the dominant factor in these vials. This was possibly due to the presence of saponin, which causes hemolysis and an increased availability of red blood cell-derived nutrients (including iron) that support the growth of bacteria.

To further improve the *Y. pestis* growth in anaerobic cultures, we interrogated the effect of supplemented Fe^+2^ and Ca^+2^. In contrast to the effect observed in the supplemented aerobic vials, no significant elevation in the bacterial counts was observed for the supplemented anaerobic cultures (Figure 6A,B). Comparing the bacterial counts obtained in the anaerobic cultures with the aerobic cultures supplemented with Fe^+2^ and Ca^+2^, we established that the latter conditions resulted in an order of magnitude increase in the bacterial counts (24 h incubation, Figure 6C,D). The calculation of the doubling time in the log-phase (Figure 6A,B) showed that the shorter doubling time characterizing the anaerobic growth was similar to that observed in the supplemented aerobic conditions (45 min). The higher total bacterial counts observed in the supplemented aerobic vs. the anaerobic conditions could reflect a shortened lag phase. Long lag phases may be expected to be more pronounced when the initial inoculum is low.

### 2.3. Accurate Identification and Antibiotic Susceptibility Testing of Y. pestis in Supplemented Blood Cultures

For plague diagnosis, blood samples of suspected infected individuals are examined for the presence of the soluble capsular antigen (F1) and V antigen, the major bacterial virulence factors [10,37,38,39]. As these biomarkers are considered to be reliable and specific for an early detection, their presence in the modified cultures was further assessed. Supplemented and non-supplemented BACTEC Aerobic Plus blood cultures were spiked with bacteria of the Kim53 or PKH-10 virulent strains. The culture samples were collected at 16–24 h incubation and the levels of F1 and V antigens were determined by an ELISA. The early detection of F1 and V was observed for both strains in the supplemented media in correlation with the accelerated growth rates promoted by metal ion supplementation. The maximal detection values (assay saturation) for F1 were quantified as early as 16 h post-inoculation in the supplemented cultures compared with the marginal values (borderline positive) detected after 20 h (with Kim53) or even undetectable levels after 24 h (PKH-10) in the non-supplemented cultures (Figure 7A,C). Surprisingly, the V antigen was detectable in the supplemented cultures after 16 h (both strains) compared with a minimal to no detection in the non-supplemented cultures after 24 h incubation (Figure 7B,D). The V antigen elevation in the Ca^+2^-supplemented culture did not align with the low calcium response mechanism of *Y. pestis.* This could possibly be explained by a differential response to the combination of Fe^+2^ and Ca^+2^ compared with Ca^+2^ alone, exerting different effects on the gene regulation [40]. This phenomenon was also determined following 24 h in the cultures of the bacteria belonging to the Alexander and IV 75 195 virulent strains (Figure 7E,F). The early detection of the F1 and V antigens directly from the blood cultures strengthened the significant improvement promoted by Fe^+2^ and Ca^+2^ supplementation as accelerators of growth for diagnostic purposes. We envisage that, in the future, using supplemented cultures may represent a means by which soluble antigen detection is implemented as a standalone assay for an even more rapid plague identification in blood culture.

Finally, we verified that supplementing the growth media did not detrimentally affect the reliability of the additional diagnostic tests either for directly identifying *Y. pestis* or for antibiotic susceptibility testing (AST). First, the performance of MALDI-TOF MS in the identification of the pathogen was evaluated. This rapid identification methodology has been proposed to represent a reliable diagnostic approach in positive blood cultures using in-house or commercial protein extraction methods [41,42]. As expected, a positive identification with high score values (>2.1) was obtained for the bacterial protein extracts derived from the supplemented blood cultures within 24 h of incubation but not from the non-supplemented bacterial extracts (which required a 40 h incubation for identification). Of note, this result suggested that the potential modification of the media did not alter the *Y. pestis* protein repertoire, allowing the MALDI-TOF identification (Figure 8A). The anti-F1 immunofluorescence staining method for a *Y. pestis* diagnosis in the blood culture samples was then tested. The results, as described in Figure 8B, established that the sensitivity of the assay actually improved, owing to the higher bacterial density afforded by the supplemented cultures (Figure 8B). Finally, the supplemented blood cultures were used as a bacterial source for AST. Several antibiotics are recommended by the Center For Disease And Prevention Control (CDC) for prophylaxis and the treatment of plague [43] including bacteriostatic (e.g., doxycycline) and bactericidal drugs (e.g., ciprofloxacin and gentamicin). Of note, a spontaneous resistance to therapeutic antibiotics rarely emerges in *Y. pestis*, yet the malicious use of this pathogen in bio-terror scenarios may be associated with the intentional engineering of strains exhibiting a resistance; therefore, the ease of isolation promoted by the modified cultures is beneficial.

Antibiogram E-test assays were performed in order to examine whether the Fe^+2^ and Ca^+2^ supplementation affected the susceptibility of the bacteria to these antibiotics compared with the non-supplemented culture. Therefore, the minimal inhibitory concentration (MIC) of *Y. pestis* derived from the supplemented compared with the non-supplemented blood cultures was determined. The lower density of the bacteria in the non-supplemented media following 24 h of incubation impaired the reliability of the test whereas in the supplemented media, the MIC was successfully determined with no differences in values compared with the standard test as the control (Figure 8C). Thus, the results clearly indicated the highly beneficial value of supplementing the cultures for the rapid diagnosis and isolation of *Y. pestis*.

## 3. Conclusions

Blood culturing represents the routine method for a bloodstream infection diagnosis. This is especially relevant considering the low specificity of the clinical presentation at the beginning of a *Y. pestis* infection. In the case of outbreaks occurring in geographic locations that do not have the benefit of well-stocked diagnostics laboratories, blood culturing may be limited in feasibility but it has an important role in molecular epidemiology surveillance and antimicrobial resistance monitoring. The conclusive identification of slow growth pathogens using blood culture methods frequently requires more than 24 h (up to 96 h). The delays in the detection and proper identification can affect the efficacy of the treatments whose early administration is critical, especially in the case of lethal pathogens such as *Yersinia pestis* [5,44]. In the present study, *Y. pestis* was representative of slow growing bacteria in vitro. In order to shorten the diagnostic time, adding external Fe^+2^ and Ca^+2^ to commercially routinely used blood culture vials was examined for the ability of enhancing the *Y. pestis* growth rates, allowing a rapid diagnosis.

The study consisted of a comparison of a variety of blood culturing conditions generated by supplemented or standard aerobic and anaerobic protocols. The data documented in this report established a significant improvement in the rate of bacterial growth in combined Fe^+2^- and Ca^+2^-supplemented media. The study suggested that the supplemented aerobic cultures represented the conditions of choice for the early detection of the pathogen both on the basis of the bacterial growth as well as the methods involving mass spectrometry, immunodetection and the quantification of the soluble bacteria-derived biomarkers.

We recommend that, in suspected plague cases, either anaerobic (as per the original protocol) or supplemented aerobic vials should be included in the sampling protocol. In such cases, positive BACTEC signals may prompt early downstream diagnostic assays for *Y. pestis* identification. Further research will establish the applicability of the metal ion supplementation for the improvement of the growth of other pathogens, possibly leading to a general universal method for accelerating the pathogen growth and identification.

## 4. Materials and Methods

### 4.1. Y. pestis Strains

Frozen bacterial stocks were prepared by growing several *Y. pestis* colonies that carried the pMT1, pCD1 or pPCP1 plasmids and the pgm locus in Tryptose Phosphate Broth (TPB, BD 260300) for 20 h in a shaking incubator (100 rpm) at 28 °C. Glycerol was added to a final concentration of 15% and the cultures were stored at −70 °C.

The study was conducted in a BSL3 facility in accordance with the biosafety guidelines of the Israel Institute for Biological Research (IIBR).

**Table 1 pathogens-11-00255-t001:** *Y. pestis* Strains.

Strain	Biovar	Ref.
Kimberley53 (Kim53)	Orientalis	[19]
Alexander	Orientalis	[11]
PKH-10	Medievalis	This study
IV 75 195	Orientalis	[11]

### 4.2. Culture Media

A 1 M solution of calcium chloride was obtained from Sigma Israel. FeSO_4_ powder (obtained from Merck) was freshly dissolved in DDW, reaching a 100 mM stock solution and was then filtered in a 0.2 µm filter. Human blood samples were obtained from the National Blood Services, MDA, Israel, under MDA research permit 08-0290.

*Y. pestis* strains were grown on BHIA (brain heart infusion agar, BD, Sparks, MD, USA) plates at 37 °C for 48 h. The colonies were suspended in sterile phosphate-buffered saline (PBS, Biological Industries, Beth Haemek, Israel) and added at a defined concentration into naïve fresh human blood. Th inoculated blood samples (10 mL/vial) were inserted into four different BACTECTM blood culture vials (Plus Aerobic/F, Standard Aerobic/F, Standard Anaerobic/F and Lytic Anaerobic/F) (BD, Sparks, MD, USA). The blood cultures were supplemented with CaCl_2_ and FeSO_4_ at a final concentration of 2.5 mM and 100 µM, respectively. The non-supplemented blood culture vials were used as a control. The inoculated blood culture vials were then shaken at 150 rpm at 37 °C in a New Brunswick Scientific C76 water bath for the indicated time periods. The initial CFU counts (time zero) were determined by plating 0.1 mL blood culture samples of serial 10-fold dilutions in duplicate; for additional time points, drop-plating was performed by plating 5–10 µL of serial 10-fold dilutions in triplicate on BHIA plates. The colony-forming unit (CFU) was determined following a 48 h incubation at 28 °C. Each experiment was conducted using at least two different human blood samples collected from distinct donors. The samples were equally distributed in 2–4 blood culture vials generating duplicates within each group per strain. A total of 12 different blood donations were used in this study.

### 4.3. MIC Determination

E-test strips (BioMerieux, Craponne, France) of selected antimicrobial agents (ciprofloxacin, doxycycline and gentamicin) were applied to an inoculated Mueller–Hinton (MH) agar surface derived from spiked blood cultures (BCs) with or without supplements. Each strip contained dried antibiotic concentration gradients that were marked with a concentration scale. The plates were incubated for 24 h at 28 °C. The MIC (minimal inhibitory concentration) values (µg/mL) were read directly from the strips according to the manufacturer’s instructions.

### 4.4. Immunofluorescence Assay (IFA) for Y. pestis Detection

The IFA assay allowed the visualization of the bacteria in different matrixes. A total of 2 µL of inoculated BC with or without supplements was applied onto slides, air dried and fixed in 10% formalin for 30 min. An Alexa 488-conjugated anti-*Y. pestis* polyclonal antibody (5 µL) [45] was applied on the spots and incubated at 37 °C for 30 min. After washing with tap water, the spots were visualized by a fluorescent microscope equipped with a green filter (excitation: 475 nm/emission: 530 nm). The bacteria were visualized in × 400 magnification using an Axioskop (Zeiss Thornwood, Thornwood, NY, USA) and a DS-iR1 camera (Nikon, Melville, NY, USA). The images were taken using NIS-elements software (Nikon, Melville, NY, USA).

### 4.5. Detection of F1 and V Antigens from Inoculated Blood Cultures by ELISA

Microtiter plates were coated with 5 µg/mL polyclonal antibodies against F1 and V antigens separately [46] overnight at 4 °C. The coated wells were blocked with 2% BSA for 2 h at 37 °C. In order to detect the F1 and V antigens, 1 mL of the inoculated BC (with or without supplements) was spun at 14,000 rpm for 5 min. A total of 50 µL from the supernatant was applied on the microtiter plates and incubated for 30 min at 37 °C following the washing step with PBS containing 0.05% Tween20. Secondary Horseradish Peroxidase (HRP)-conjugated polyclonal anti-F1 or V antibodies were added and the plate was incubated at 37 °C for 30 min. Following an additional wash step, 50 µL of tetramethylbenzidine (TMB, Sigma, Jerusalem, Israel) was added to each well and the plate was incubated at room temperature for 20 min. The absorbance was measured at a 630 nm wavelength.

### 4.6. Bacterial Identification by MALDI-TOF MS

A direct bacterial identification using an MBT Sepsityper^®^ IVD kit (Bruker Daltonics GmbH, Bremen, Germany) was performed according to the manufacturer’s instructions. Briefly, 1 mL of blood culture fluid was collected and transferred into a 1.5 mL tube followed by the addition of a 200 µL lysis buffer. The mix was centrifuged for 2 min at 14,000 rpm and the pellet was washed with a 1 mL washing buffer. The mix was re-centrifuged for 1 min at 14,000 rpm. The supernatant was discarded and the pellet was dried. The proteins of the sample were extracted using a standard EX method according to the manufacturer’s instructions. The samples were then identified by a Bruker MALDI-TOF MS instrument (Bruker Daltonics, GmbH, Bremen, Germany).

### 4.7. Statistical Analysis

The data were analyzed using GraphPad Prism5 software. The results were expressed as a mean ± standard error. The statistical significance was determined by a Student’s *t*-test and a two-way ANOVA. A *p*-value ≤ 0.05 was considered to be significant.

## Figures and Tables

**Figure 1 pathogens-11-00255-f001:**
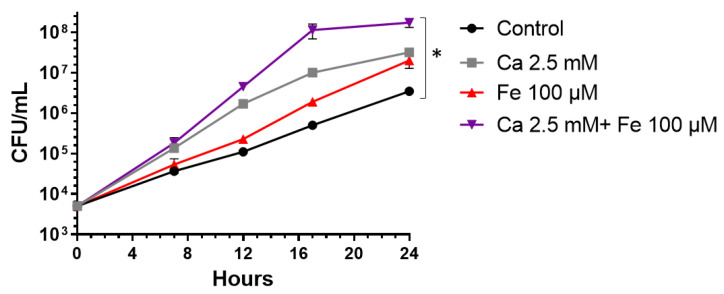
The effect of iron and calcium supplementation on *Y. pestis* growth rates. BACTEC Plus Aerobic/F culture vials containing 10 mL of naïve fresh human blood were spiked with the virulent *Y. pestis* Kimberley53 (Kim53) strain at a final concentration of 5 × 10^3^ CFU/mL. Fe^+2^ (100 µM) and Ca^+2^ (2.5 mM), separately or in combination, were added to vials and then incubated at 37 °C. Bacterial growth was determined 7, 12, 17 and 24 h following incubation by CFU counts. Non-supplemented blood cultures were used as controls. Results are averages ± SEM of four blood cultures for each group containing two individual blood donations and representing two independent experiments obtaining similar results. * *p* < 0.01 of Fe^+2^- and Ca^+2^-supplemented culture vs. non-supplemented control according to a two-tailed Student’s *t*-test.

**Figure 2 pathogens-11-00255-f002:**
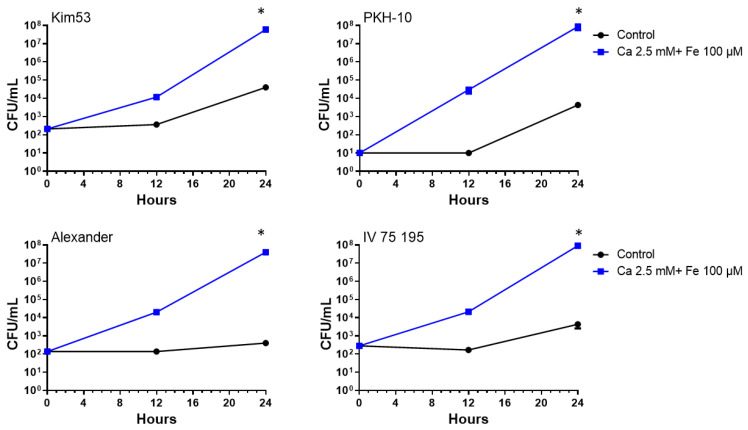
Accelerated growth rate of *Y. pestis* virulent strains in supplemented blood culture. BACTEC Plus Aerobic/F culture vials containing 10 mL of naive fresh human blood were spiked with four different virulent *Y. pestis* strains (Kim53, PKH-10, Alexander and IV 75 195) at a final concentration of 10–500 CFU/mL. Vials were supplemented with Fe^+2^ (100 µM) and Ca^+2^ (2.5 mM) and then incubated at 37 °C. Non-supplemented blood cultures were used as a control. Bacterial growth was determined following 12 h and 24 h incubation by CFU counts. Results are averages ± SEM of three counts each from two blood cultures per group. * *p* < 0.01 of supplemented vs. non-supplemented control culture according to a two-tailed Student’s *t*-test.

**Figure 3 pathogens-11-00255-f003:**
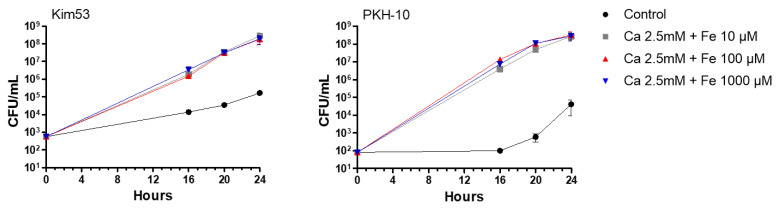
Optimization of iron concentrations for enhancing bacterial growth. BACTEC Plus Aerobic/F culture vials containing 10 mL of naive fresh human blood were spiked with Kim53 (500 CFU/mL) and PKH-10 (100 CFU/mL). Vials were supplemented with Ca^+2^ (2.5 mM) combined with three different concentrations of Fe^+2^ (10, 100 and 1000 µM). Non-supplemented blood cultures were used as a control. Vials were incubated at 37 °C and bacterial growth was determined following 16, 20 and 24 h incubation by CFU counts. Results are averages ± SEM of three counts each of duplicate blood cultures containing two individual blood donations in each group.

**Figure 4 pathogens-11-00255-f004:**
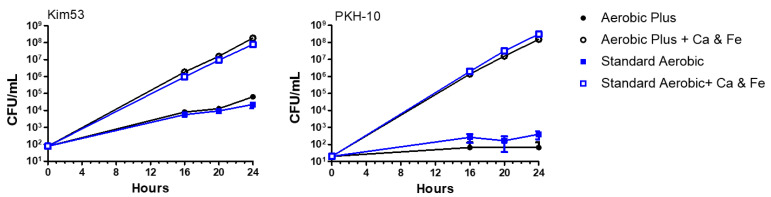
Enhanced bacterial growth in supplemented blood cultures is maintained in BACTEC Standard Aerobic vials. BACTEC Standard Aerobic/F and BACTEC Plus Aerobic/F vials containing 10 mL of naive fresh human blood were spiked with Kim53 (100 CFU/mL) and PKH-10 (10 CFU/mL). Vials were supplemented with 2.5 mM Ca^+2^ and 100 µM Fe^+2^. Non-supplemented blood cultures were used as a control. Vials were incubated at 37 °C and bacterial growth was determined following 16, 20 and 24 h incubation by CFU counts. Results are averages ± SEM of three counts each of duplicate blood cultures containing two individual blood donations in each group.

**Figure 5 pathogens-11-00255-f005:**
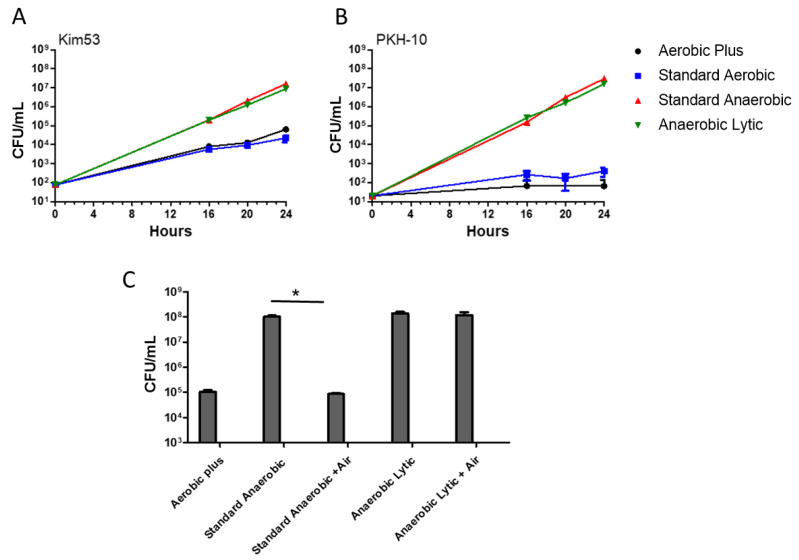
Superior bacterial growth in anaerobic vs. aerobic vials. Aerobic and Anaerobic BACTEC vials (Standard Aerobic, Plus Aerobic, Standard Anaerobic and Anaerobic Lytic) containing 10 mL of naive fresh human blood were spiked with Kim53 (100 CFU/mL) and PKH-10 (10 CFU/mL). Vials were incubated at 37 °C and bacterial growth was determined following 16, 20 and 24 h incubation by CFU counts. A comparison of the natural growth of Kim53 (**A**) and PKH-10 (**B**) under aerobic and anaerobic conditions was evaluated. * *p* < 0.01 of anaerobic vs. aerobic vials according to a two-way ANOVA test. (**C**) The role of anaerobic conditions in enhancing *Y. pestis* growth. Anaerobic BACTEC vials (Standard and Lytic) containing 10 mL of naive fresh human blood were spiked with Kim53 (100 CFU/mL). The role of the anaerobic condition in bacterial growth was determined by inserting air into the vials, replacing the N_2_/CO_2_ atmosphere prior to incubation compared with the growth obtained in non-aerated vials. BACTEC Aerobic Plus blood culture was used as a control. Results are averages ± SEM of three counts of duplicate blood cultures containing two individual blood donations in each group. * *p* < 0.01 of standard anaerobic vs. anaerobic + air vials according to a two-tailed Student’s *t*-test.

**Figure 6 pathogens-11-00255-f006:**
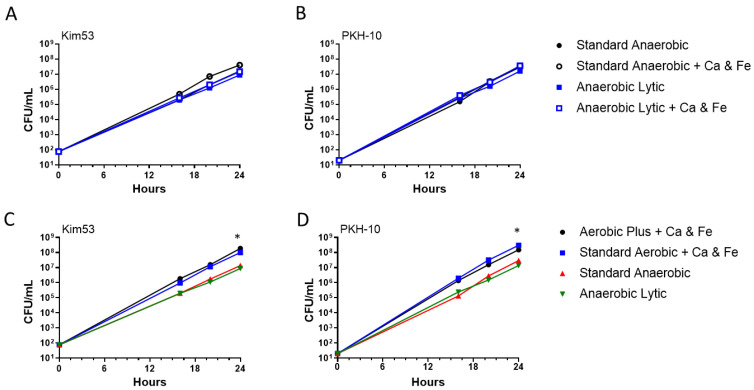
Superior bacterial growth in supplemented aerobic vs. anaerobic vials. Aerobic and Anaerobic BACTEC vials (Standard Aerobic, Plus Aerobic, Standard Anaerobic and Anaerobic Lytic) containing 10 mL of naive fresh human blood were spiked with Kim53 (100 CFU/mL) and PKH-10 (10 CFU/mL). Vials were incubated at 37 °C and bacterial growth was determined following 16, 20 and 24 h incubation by CFU counts. (**A**,**B**) The role of supplemented Ca^+2^ (2.5 mM) and Fe^+2^ (100 µM) in anaerobic cultures was evaluated. Non-supplemented anaerobic vials were used as a control. (**C**,**D**) A superior increase in bacterial counts was achieved in aerobic supplemented vs. anaerobic cultures. Results are averages ± SEM of three counts each of duplicate blood cultures containing two individual blood donations in each group and represent two independent experiments obtaining similar results. * *p* < 0.01 of supplemented aerobic vs. non-supplemented anaerobic vials according to a two-way ANOVA test.

**Figure 7 pathogens-11-00255-f007:**
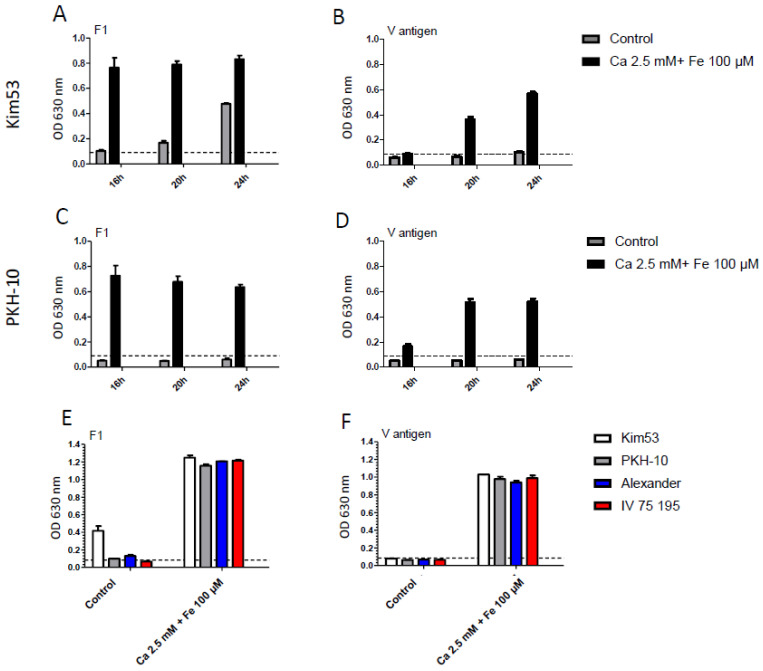
Detection of plague virulence biomarkers V and F1 antigens from inoculated blood culture. Blood cultures (BACTEC Plus Aerobic/F culture vials) containing 10 mL of naive fresh human blood were spiked with (**A**,**B**) Kim53 (500 CFU/mL) and (**C**,**D**) PKH-10 (100 CFU/mL). Ca^+2^ (2.5 mM) and Fe^+2^ (100 µM) were supplemented into vials. Non-supplemented blood cultures were used as a control. Vials were incubated at 37 °C and blood cultures were sampled following 16, 20 and 24 h incubation for the detection of plague biomarkers F1 and V antigens by ELISA. (**E**,**F**) F1 and V antigen detection following 24 h incubation of blood culture inoculated with four different virulent *Y. pestis* strains (Kimberley53, PKH-10, Alexander and IV 75 195) at a final concentration of 10–500 CFU/mL. Black dashed line represents OD 630 nm 0.1, the limit of detection of the test. Results are averages ± SEM of triplicates of two blood cultures in each group containing two individual blood donations.

**Figure 8 pathogens-11-00255-f008:**
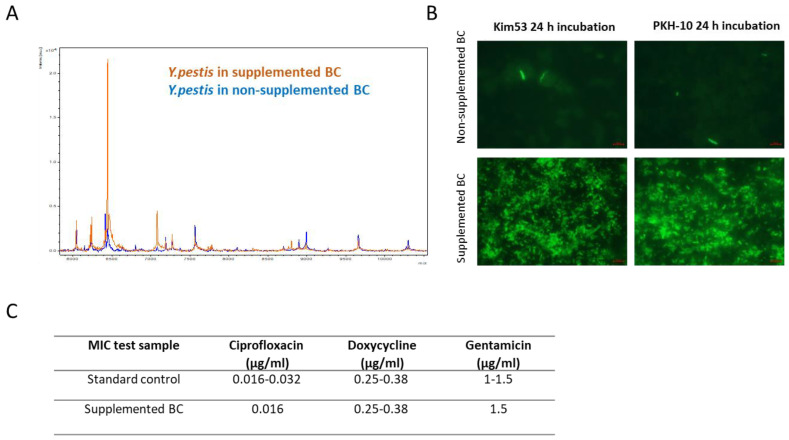
Direct detection and susceptibility assays for *Y. pestis* in supplemented blood cultures. BACTEC Plus Aerobic/F culture vials containing 10 mL of naive fresh human blood were spiked with Kim53 and PKH-10. Ca^+2^ (2.5 mM) and Fe^+2^ (100 µM) were supplemented into vials and non-supplemented blood cultures (BC) were used as a control. (**A**) MALDI-TOF analysis of Kim53 bacterial protein extraction derived directly from supplemented and non-supplemented cultures following 24 h and 40 h incubation, respectively. Spectrum obtained from a protein extract derived from supplemented culture completely overlapped with the spectrum obtained from non-supplemented control culture (orange and blue spectra, respectively). (**B**) Immunofluorescence assay (IFA) using A488-conjugated anti-*Y. pestis* F1 polyclonal Abs for the detection of the bacteria directly from blood culture following 24 h incubation. (**C**) E-test assays to determine the MIC (µg/mL) of ciprofloxacin, doxycycline and gentamicin were performed directly from supplemented blood cultures (following 24 h incubation) and compared with standard tests (performed using bacteria suspended in PBS and plated).

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
