# Peer review of "An Improvement in Diagnostic Blood Culture Conditions Allows for the Rapid Detection and Isolation of the Slow Growing Pathogen Yersinia pestis"

_pathogens, 2022, doi:10.3390/pathogens11020255_

Round 1
Reviewer 1 Report
Thank you for providing me the opportunity to review this interesting piece of work. From the microbiological perspective, I a have very few to ad to the material and methods of the study. Material and methods are soundly and support the notion that aerobic/anaerobic supplemented culture with Fe and C. The superiority of enriched aerobic media is well stated. I have just a question about the number of tests carried out. This is not explained how many bottles have been inoculated for each strain and then, the variability of the results could not be evaluated.
In second term, I suggest to ad in the discussion the impact of these findings in real world. Most of plague cases in the world currently arise in rural and remote areas from Madagascar and the Ituri region of Democratic Republic of Congo, besides limited outbreaks or sporadic cases in other countries. The availability of blood culture in those settings is extremely scarce if not unavailable and made these procedures with limited utility to respond on time to an outbreak or suspected case of plague. With this sense, immuno-crhomatography from samples obtained from suspected buboes (bubonic plague is by large the most frequent clinical presentation) remain the most useful diagnostic tool in these contexts. RT-PCR methods may provide faster and easier way for early diagnostic of plague than blood culture, except for pneumonic and septic presentation.
Most probably, blood culture may have a major role on diagnosis of pneumonic plague, due to the low specificity of the clinical presentation at the beginning of symptoms and the low yield of immuno-crhomatography on sputum samples, and the systematic sepsis associated to this clinical presentation.
Finally, it should be mentioned that antibiotic resistance is, fortunately, a very rare finding on Y. pestis strains, but this do not exclude the necessity of regularly testing all samples isolated. Blood culture would have an important role on strains isolation and therefore, molecular epidemiology studies and antimicrobial resistance monitoring, but not as a tool for outbreak response.
I suggest to discuss these issues to complement the impact of the findings.
Author Response
Reviewer #1
Thank you for providing me the opportunity to review this interesting piece of work. From the microbiological perspective, I a have very few to ad to the material and methods of the study. Material and methods are soundly and support the notion that aerobic/anaerobic supplemented culture with Fe and C. The superiority of enriched aerobic media is well stated. I have just a question about the number of tests carried out. This is not explained how many bottles have been inoculated for each strain and then, the variability of the results could not be evaluated.
Author response: We thank the reviewer and his favorable assessment of the study. The requested information has been incorporated in the revised version of the manuscript in the Materials and Methods section (Lines 418-421).
In second term, I suggest to ad in the discussion the impact of these findings in real world. Most of plague cases in the world currently arise in rural and remote areas from Madagascar and the Ituri region of Democratic Republic of Congo, besides limited outbreaks or sporadic cases in other countries. The availability of blood culture in those settings is extremely scarce if not unavailable and made these procedures with limited utility to respond on time to an outbreak or suspected case of plague. With this sense, immuno-crhomatography from samples obtained from suspected buboes (bubonic plague is by large the most frequent clinical presentation) remain the most useful diagnostic tool in these contexts. RT-PCR methods may provide faster and easier way for early diagnostic of plague than blood culture, except for pneumonic and septic presentation.
Most probably, blood culture may have a major role on diagnosis of pneumonic plague, due to the low specificity of the clinical presentation at the beginning of symptoms and the low yield of immuno-crhomatography on sputum samples, and the systematic sepsis associated to this clinical presentation.
Finally, it should be mentioned that antibiotic resistance is, fortunately, a very rare finding on Y. pestis strains, but this do not exclude the necessity of regularly testing all samples isolated. Blood culture would have an important role on strains isolation and therefore, molecular epidemiology studies and antimicrobial resistance monitoring, but not as a tool for outbreak response.
I suggest to discuss these issues to complement the impact of the findings.
Author response: These suggested valuable comments were added to the revised manuscript (Lines 333-337 and 366-370 )

Reviewer 2 Report
In the manuscript ID: pathogens-1592816, the authors describe the effect of supplementing blood cultures with iron and calcium in the detection and isolation of Yersinia pestis in blood samples. Applying a final concentration of 2.5 and 100 µM for calcium and iron, respectively, resulted in the pathogen accelerated growth rate, in comparison to standard culture conditions, a behaviour that resembles the growth in anaerobic cultures; moreover, even the detection of the capsular F1 and V antigen resulted improved in presence of the nutritional supplementation, which, on the other hand, did not impair antibiotic susceptibility tests. Such results were further confirmed by MALDI-TOF identification and anti-F1 immunofluorescence staining.
The paper is very interesting, well-organized and easy to understand. The topic has been well introduced, the experimental data are clearly reported and well commented, with all the proper controls and the related statistical significance.
There are just minor modifications, mainly in the methodological section, to be applied:
-Please introduce the bacterial strains table with a proper sentence and report the conditions adopted for strains conservation.
-Please specify magnification and fluorescent filters used in Y. pestis immunofluorescence detection.
-Please specify the applied statistical analysis methods.
Moreover, about the putative adoption of supplemented cultures in diagnostics, did the authors consider their effect on further potential pathogens in blood samples? Could this modification impact on bacterial growth and somehow hamper the detection and isolation of Y. pestis? Please comment.
After these minor revisions, the manuscript can be published in “Pathogens”.
MINOR COMMENTS
Please indicate correctly the iron and calcium ions (ex Ca2+) throughout the manuscript;
Please type Y. pestis in italic throughout the manuscript;
Line 43, please use the full name “Yersinia pestis” and correct “Gram-negative”;
Line 332, please correct “supplementation affects”;
Author Response
Reviewer #2
In the manuscript ID: pathogens-1592816, the authors describe the effect of supplementing blood cultures with iron and calcium in the detection and isolation of Yersinia pestis in blood samples. Applying a final concentration of 2.5 and 100 µM for calcium and iron, respectively, resulted in the pathogen accelerated growth rate, in comparison to standard culture conditions, a behaviour that resembles the growth in anaerobic cultures; moreover, even the detection of the capsular F1 and V antigen resulted improved in presence of the nutritional supplementation, which, on the other hand, did not impair antibiotic susceptibility tests. Such results were further confirmed by MALDI-TOF identification and anti-F1 immunofluorescence staining.
The paper is very interesting, well-organized and easy to understand. The topic has been well introduced, the experimental data are clearly reported and well commented, with all the proper controls and the related statistical significance.
Author response: We thank the reviewer and his favorable assessment of the study.
There are just minor modifications, mainly in the methodological section, to be applied:
-Please introduce the bacterial strains table with a proper sentence and report the conditions adopted for strains conservation. Author response: The requested information has been incorporated in the revised version of the manuscript (Lines 394-397).
-Please specify magnification and fluorescent filters used in Y. pestis immunofluorescence detection.
Author response: The requested information has been incorporated in the revised version of the manuscript (Lines 431-440).
-Please specify the applied statistical analysis methods.
Author response: According to the reviewer comment we have now introduced data detailing the statistical analysis in the Material and Method section (line 463-467)
Moreover, about the putative adoption of supplemented cultures in diagnostics, did the authors consider their effect on further potential pathogens in blood samples? Could this modification impact on bacterial growth and somehow hamper the detection and isolation of Y. pestis? Please comment.
Authors response: The reviewer is obviously correct in pointing out that the effect of Ca and Fe in the outcome blood culture may be compromised by the presence of an additional bacterial species. In the overwhelming majority of cases, the blood cultures are homogeneous namely, there is only one species generating the bacteremic state of the patient. The presence of a second species most likely would originate from contamination of the culture which may have occur in the course or post-collection. Such occurrences are indeed pretty frequent yet the vast majority of contaminations are due to the presence of skin-borne microorganisms which are inherently fast growing and do not require Ca an Iron enrichment for growth. Furthermore, in case of contaminations, in clinical set-ups, the culture is compromised and discarded as irrelevant. Therefore, with all due respect, we do not believe that this issue should be elaborated in the manuscript.
After these minor revisions, the manuscript can be published in “Pathogens”.
MINOR COMMENTS
Please indicate correctly the iron and calcium ions (ex Ca2+) throughout the manuscript;
As requested by the reviewer, “Iron” and “calcium” ions were replaced with Fe+2 and Ca+2 at the relevant indications throughout the manuscript
Please type Y. pestis in italic throughout the manuscript; We changed it accordingly
Line 43, please use the full name “Yersinia pestis” and correct “Gram-negative”; We changed it accordingly
Line 332, please correct “supplementation affects”; We changed it accordingly
